# Vectorial Graph Convolutional Network

## Abstract

Graph Convolutional Networks (GCN) have drawn considerable attention recently due to their outstanding performance in processing graph-structured data. However, GCNs still limited to the undirected graph because they theoretically require a symmetric matrix as the basis for the Laplacian transform. This causes the isotropic problem of the operator and reduced sensitivity in response to different information. In order to solve the problem, we generalize the spectral convolution operator to directed graphs by field extension, which improves the edge representations from scalars to vectors. Therefore, it brings in the concept of direction. That is to say, and even homogeneous information can become distinguishable by its differences in directions. In this paper, we propose the Vectorial Graph Convolutional Network(VecGCN) and the experimental evidence showing the advantages of a variety of directed graph node classification and link prediction tasks.

## 1 Introduction

A graph is a ubiquitous data structure where entities are vertices and edges are their pairwise relationships. Most Graph Neural Networks(GNNs) fall into one of two categories: spectral Defferrard et al. (2016); Kipf & Welling (2016) or spatial networks Hamilton et al. (2017a); Veličković et al. (2017); Backstrom & Leskovec (2011). Spatial approaches are based on a localized averaging operator with learnable weights that iteratively traverse the entire graph. Spectral approaches based on eigen-decomposition of graph Laplacian and smooth those signals through Fourier transform Zhou et al. (2020); Wu et al. (2020). The application domains ranging from social networks Chen et al. (2012) to quantum chemistry Liao et al. (2019) and text classification Yao et al. (2019), etc. One of the key techniques is Graph Convolutional Networks (GCNs) Defferrard et al. (2016); Kipf & Welling (2016); Xu et al. (2018a), it's the variant of Convolutional Neural Networks (CNNs) Mallat (2016) on graphs, that learns the representations from both vertices and edges. It is particularly important to apply representations to downstream tasks Hamilton et al. (2017b), e.g., node classification and link prediction Hu et al. (2020).

However, the vast majority of these researches are based on undirected graphs, and even the original graphs are naturally directed. This phenomenon will take the risk of discarding potentially important information Kawamoto et al. (2018); Zhang et al. (2021). For example, you may have heard of a celebrity, but he/she doesn't know you. From the GATs' perspective, it is easy to understand that the attention values from node $i$ to node $j$ and node $j$ to node $i$ are not necessarily equal, which means the information is not symmetric on the edges.

Adjacency matrix $A$ is the topological edge set. Unless graph $G$ is undirected, $A$ is not symmetric. Unfortunately, GCNs are developed from spectral theory Kipf & Welling (2016); Xu et al. (2018a); Gilmer et al. (2017) and limited to symmetric convolutional kernels Beaini et al. (2021), the object matrix of the kernels needs to be positive semi-definite and symmetric because the decomposition of a such matrix is orthogonal that can be taken as Fourier transform basis. It, in turn, requires the graph to be undirected to satisfy the above two conditions, or the eigenvalues of $A$ can not be solved in the real number field. Thus, extending spectral methods to directed graphs is not straightforward Zhang et al. (2021).

Therefore, one of the key challenges lies in defining a symmetric adjacency matrix on a directed graph.

Recently, there has been a surge of interest in directed GCNs Tong et al. (2020b;a); Monti et al. (2018); Beaini et al. (2021); Zhang et al. (2021). These studies have proposed different approaches to solve the problem. However, the original purpose of constructing directed GCNs is to keep more information from the graph. While these previous studies view the direction as a one-dimensional scalar, it is supposed to be a vector that shares the same dimension with the node vector. From a Principal Component Analysis(PCA) perspective Abdi & Williams (2010), The $n$-dimensional node vectors is decomposed into $k$ principal components ($k \leq n$), GCNs preserved the $1st$ component on edge and directed GCNs preserved the $1st$ and $2st$ components. This shows that some of the information is lost.

To address these issues, we proposed VecGCN. We overcome the symmetric problem and the information loss problem at the same time by using Field Extension, which is the main research object of field theory in abstract algebra. The basic idea is to start from a base field and somehow construct a "larger" field that contains it. And we construct a high dimensional field according to distances between nodes. Firstly, the distance matrix is symmetric. Secondly, a high dimensional field does not cause information loss.

The main contributions are summarized as follows:

1. **Replace the adjacency matrix with the distance matrix.** The advantages of replacing the adjacency matrix with the distance matrix include two main aspects. On the one hand, the distance matrix is symmetric. On the other hand, the topology of the distance matrix is the same as that of the adjacency matrix. If the distance between two adjacent nodes is 1, then the distance matrix and the adjacency matrix are equal. Therefore, we can consider the distance matrix as a generalization of the adjacency matrix. Not only that, but the distance matrix is also satisfied by the theory of GCN.

2. **The concept of direction is proposed.**
   Since the adjacency matrix is binary, 0 indicates that there is no edge between two nodes, while 1 is the opposite. Therefore the adjacency matrix does not imply the concept of direction, which makes the model unable to distinguish the importance of neighboring nodes, which manifests isotropy. The above problem can be solved by improving the elements of the adjacency matrix from scalars to vectors using the field extension method.

3. **Propose the VecGCN.** Our extensive experiments on a series of datasets clearly show that VecGCN's performance exceeds most other methods.

## 2 RELATED WORK

Most graph neural network structures can be categorized as either spectral or spatial. Neighborhoods in spatial networks such as Veličković et al. (2017); Hamilton et al. (2017a); Atwood & Towsley (2016); Duvenaud et al. (2015) are well-defined even when their adjacency matrices are not symmetric. Although spatial methods typically have natural extensions to directed graphs, they may ignore important information in the directed graph, as we discussed before. Spectral approaches also suffer from this problem. In this section, we review related work on constructing neural networks for directed graphs, and describe the development of the problem in detail as well as the various solutions. We refer the reader to Wu et al. (2020); Zhang et al. (2020) for more background information.

### 2.1 NOTATIONS AND PRELIMINARIES.

Given a simple and connected undirected graph $G = (V, E)$ with $n$ nodes and $m$ edges. Let $A$ denote the adjacency matrix and $D$ the diagonal degree matrix. Spectral-based GCNs are based on the Laplacian matrix, and the graph Laplacian matrix is defined as $L = D - A$. The normalized format of Laplacian matrix is defined as $L_{sym} = D^{-\frac{1}{2}} L D^{-\frac{1}{2}} = I - D^{-\frac{1}{2}} A D^{-\frac{1}{2}}$, where $I$ is an identity matrix that has same shape with $A$. $L_{sym}$ is a matrix representation

of a graph in graph theory and can be used to find many useful properties. It is a symmetric positive semi-definite matrix. With these properties, the eigen-decomposition of normalized Laplacian matrix $L_{sys}$ write as $U\Lambda U^T$. Here $\Lambda$ is a diagonal matrix of the eigenvalues of $L_{sys}$, and $U \in \mathcal{R}^{n \times n}$ is a unitary matrix that consists of the eigenvectors of $L_{sys}$. Let $X \in \mathcal{R}^{n \times d}$ denote the representation of node feature matrix, that is, each node in $l-th$ layer is associated with a $d$-dimensional feature vector $X^l = [x_1^l, \cdots, x_n^l]$. While $h$ is the input node feature and is only used in the input layer. The graph convolution operation between signal is defined as $g_\theta(L) * x = U g_\theta(\Lambda) U^T x$, where the feature vector $x \in X$ is the signal and $g_\theta(\Lambda)$ the spectral filter, and $\theta$ corresponds to a vector of spectral filter coefficients. Finally, Each hidden layer $l$ is assigned with learnable parameters $W^l$, and $\sigma(\cdot)$ is an activation function such as ReLU, and $\langle \cdot, \cdot \rangle$ represents the inner product between any two vectors.

## 2.2 Vanilla GCNs

GCN is a generalization of the Fourier transform on the graph. Bruna et al. (2013) proposed first generation of GCN, written as:

$$X^{l+1} = \sigma(U g_\theta(\Lambda) U^T X^l)$$

Since the computational complexity is $\mathcal{O}(n^3)$, Defferrard et al. (2016) (referred to as Cheb-Net) proposed the second generation of GCN, and model outputs can be approximated by the $K - th$ order polynomial of Laplacians by Chebyshev Polynomials, written as:

$$X^{l+1} = \sigma(\sum_{i=0}^{K} \alpha_i L_i X^l)$$

The derivation can be found in B.1. Kipf & Welling (2016) further simplified the model, they set $K = 1$ and use the renormalization trick, replaces $L_{sys}$ by a normalized version $\widetilde{L} = \widetilde{D}^{-\frac{1}{2}} \widetilde{A} \widetilde{D}^{-\frac{1}{2}} = (D+I)^{-\frac{1}{2}} (A+I)(D+I)^{-\frac{1}{2}}$, obtain:

$$X^{l+1} = \sigma(\widetilde{L} X^l W^l)$$

According to the above equation, we know that the vanilla GCN is a multi-layer neural network that receives and update node features across the graph.

## 2.3 Limitition of undirected GCNs

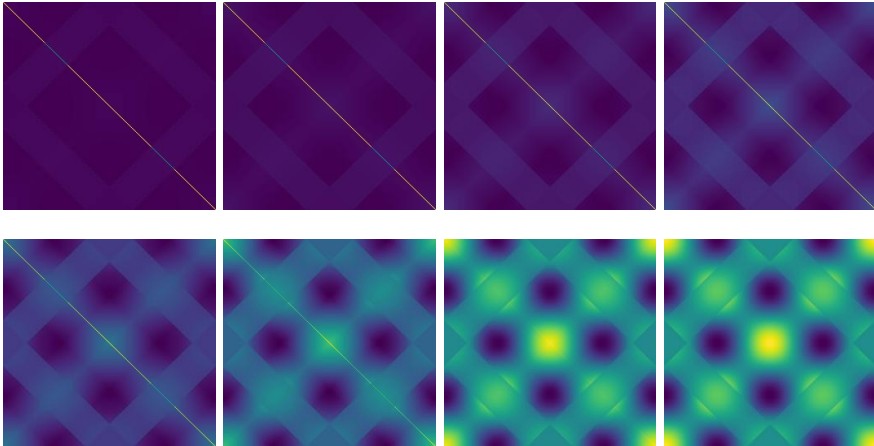

Figure 1: Examples of ChebNet's outputs on a circulant graph. Note that the graph convolutional operators are isotropic on some local areas due to rotational invariance. Hence, it is difficult to define directions.

Since $\widetilde{L}$ is a symmetric matrix, spectral-based GCNs are limited to applying to undirected graphs Wu et al. (2020). Unlike convolutional neural networks, GCNs' convolutional operators are symmetric. One way to apply GCNs to a directed graph is to relax the graph

structure by trivially adding edges to symmetrize the adjacency matrices Kipf & Welling (2016); Wu et al. (2019). However, it loses the distinctive structure of the directed graph and misleads the message passing scheme with incorrect weights Wang et al. (2020). Almost all relationships in nature are irreversible, such as time-series relationships, parent-child relationships, virus transmission relationships, and thermal conductivity relationships. Besides, it is not capable to learn from complex local structureMonti et al. (2018), inheritance relationship Kampffmeyer et al. (2019) and second-order proximity Tong et al. (2020b). The above problems are mainly due to the lack of anisotropy in the GCN convolution operator of GCNMonti et al. (2018); Beaini et al. (2021) as shown in Fig~1.

## 2.4 THREE TYPES OF EDGE VECTORS

To the best of our knowledge, the edge direction in GNNs are divided into four types:

1. **Direction of information flow.** Information flow is a series of samples of nodes. Since GNNs are based on the Message Passing(MP) techniqueGilmer et al. (2017), that allows nodes repeatedly aggregate and update their own representations. Different sampling methods cause MP to produce different aggregation results. Therefore, these sampling methods generate different information flows and significantly affect the models' performances Tong et al. (2020b).

2. **Edge embedding and representation.** Many graph data in the real world contain multiple types of nodes and edges, and these graphs are called heterogeneous graphs Gasteiger et al. (2019). Typical heterogeneous graphs include molecular graphs, academic graphs, and recommendation graphs. In order to distinguish different types of edge information, embedding methods are used to represent different edges Wang et al. (2021); Schlichtkrull et al. (2018); Zhang et al. (2019). This approach brings in additional learnable modules based on the GCN model. Further, another approach Pei et al. (2020); Klicpera et al. (2020) constructs a continuous space underlying the graph by combining different types of edge representations.

3. **Spectral direction.** By constructing the Hermitian Laplacian matrix Zhang et al. (2021), the complex eigenvalues are generated after decomposition. The real part of an eigenvalue indicates the presence of an edge, and the imaginary part indicates the direction of the edge. Since the obtained directions are unlearnable scalars, it is also necessary to bring in learnable modules to improve the model's performance.

4. **Spatial direction.** The Spatial approach is the most intuitive, where the edge vector is defined as the difference between two adjacent nodes Beaini et al. (2021), $\vec{e}_{ij} = u_i - u_j$ where $i \neq j$.

Our proposed VecGCN belongs to the Spatial approach. The advantage of this approach is that it is very natural to understand the meaning of the direction, the same as our intuition in daily life. In addition, the design of VecGCN is simple and hardly changes the structure of vanilla GCN.

Based on vanilla GCN, we define the vector field on the graph and prove the generalization of the method on an n-dimensional grid, which provides the basis for subsequent analysis. The vector in the vector field is divided into direction and length. The difference between the two nodes defines the direction. Inspired by GATVeličković et al. (2017), we define the length as the importance of an edge, so we again design a normalization function to calculate the significance of each edge.

## 3 METHOD

Since GCNs are based on the explicit assumption of an undirected graph, this leads to a drawback that the Laplacian operator is rotationally symmetricB.5. For example, the order of neighbor nodes does not change the final aggregation result, which is isotropicMonti et al. (2018), making it no preferred direction on the graph. In order to solve the above problems, we propose the VecGCN, and the key idea is to expand the definition domain of the adjacency matrix using the field expansion method.

### 3.1 Permutation invariance of distance function

It is not easy to keep the symmetry of the adjacency matrix. However, we notice that the distance is the same between arbitrary two adjacency node $i$ and node $j$, written as $d(i,j) = d(j,i)$, where $d$ is a $L^2 - norm$ distance function defines in Euclidean space, it tells us the distance function is permutation invariant. What is more, the distance matrix is symmetric. Moreover, the topology of the distance matrix and the adjacency matrix are the same because the adjacency nodes define both. Therefore, in this paper, we replace the adjacency matrix with the distance matrix and note it as $\mathcal{F}$, which is the first step in constructing VecGCN.

### 3.2 Field Extension

With the help of Field ExtensionB.4 , distance matrix can be extend to high dimension, $\mathcal{R} \to \mathcal{R}^n$. In this paper, we set the distance matrix to 64-dimension, note as $F^{64}$ and $[F^{64}/F] = 64$. Since $\mathcal{F}$ is the basis of $\mathcal{F}^{64}$, we can view $\mathcal{F}^{64}$ as a linear combination of $\mathcal{F}$, more details are provided in Appendix B.2. That is to say, we can keep a symmetric distance matrix in high dimensions. This is the second step in constructing VecGCN.

### 3.3 VecGCN

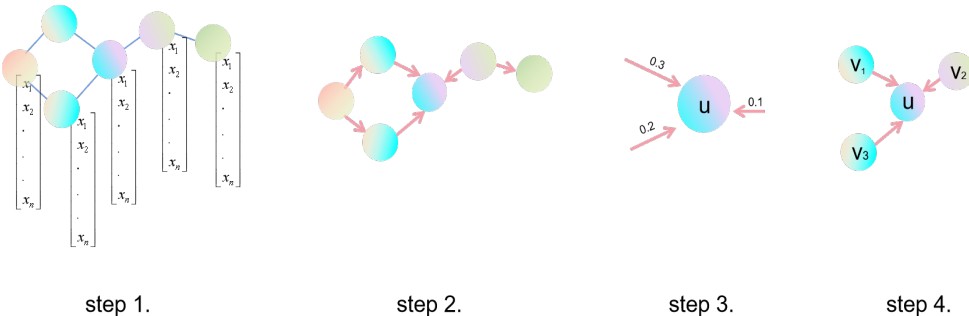

step 1.                step 2.                step 3.                step 4.

Figure 2: The four steps of VecGCN are to calculate the node representation, the edge direction, the local importance degree of each edge, and update the node representation, respectively.

Now we have a high dimensional form of distance matrix $\mathcal{F}^{64}$ to replace the adjacent matrix in vanilla GCNs. Overall speaking, in order to adapt the distance matrix $\mathcal{F}^{64}$, we also need to upgrade both the learnable parameter $W$ and the input node feature $h$ to 64 dimensions, in which the learnable parameter $W$ is easy to modify by adding only one additional dimension, and the modification of the input feature $h$ requires a linear layer mapping it to 64-dimension. In addition, while keeping the structure of GCN unchanged, it is necessary to use the partition function to normalize edge vectors.

First, a linear transformation is applied to the node feature to map it to a higher dimensional space Eq~1.

$$X_0 = \phi(h) + b \tag{1}$$

Where $\phi$ is a linear transform that operates on the input feature space, and $b$ is a basis term[1].

Second, calculate the edge vectors.

$$e_{ij}^l = x_i - x_j \tag{2}$$

$e_{ij}^l$ is defined on the Euclidean space, it is natural and easy to understand the definition of an edge vector in terms of the difference between two adjacent nodes. The superscript $l$

---

[1]The $\phi$ and the bias term are only included in this section and will be omitted in later sections.

in Eq~B.10 represents the $l$th layer. Besides, the distance is shift-invariant, and its visual explanation can be found in A1.

Third, compute the length of $e_{ij}^l$, which is the $L^2 - norm$ $||e_{ij}^l||_2$, it indicates the strength of the interrelationship between node $i$ and $j$. Then normalize the vector length with the partition function $Z_{ij}$ in Eq~3 so that $||e_{ij}^l||_2 \in [0, 1]$. Basically, $||e_{ij}^l||_2$ is expected to be small if node $i$ and $j$ are close or similar to each other.

$$Z_{ij} = \sum_{j \in \mathcal{N}(i)} ||e_{ij}^l||_2 \tag{3}$$

Fourth, $\alpha_{ij}^l$ in Eq~4 is the importance coefficient that measures the relatedness between node $i$ to node $j$ with respect to layer $l$, and it is inversely proportional to the length of the edge vector. $D_i$ is a smoothing value that indicates the degree of node $i$.

$$\alpha_{ij}^l = (1 - \frac{||e_{ij}^l||_2}{Z_{ij}})/D_i \tag{4}$$

Due to the value domain of $\alpha$ is in $[0, 1 - 1/D_i]$B.3, we know that $\alpha \in [0, 1]$, it tells us that the smaller the degree of a node, the less significant it is, i.e., a node with a minor degree can be ignored.

Subsequently, according to Eq~3 and Eq~4, the new edge vector can be updated by $e_{ij}^{l+1} = \alpha_{ij}^l \times e_{ij}^l$. For example, in Fig~2, the lengths of the three adjacent edges of node $u$ are 0.3, 0.2, 0.1. These edges' lengths after normalization Eq~3 equals 0.50, 0.33, 0.16 and corresponding importance coefficients are 0.17, 0.11, 0.05.

Fifth, under the message passing scheme, the update of the node vector can be determined by Eq~5.

$$x_i^{l+1} = \sum_{j \in \mathcal{N}(i)} \alpha_{ij}^l e_{ij}^l x_{ij}^l \tag{5}$$

In summary, as shown in Eq~6, the way each layer is updated can be written as:

$$X^{l+1} = \sigma(\widetilde{L} \alpha^l e^l X^l W^l) \tag{6}$$

From Eq~6 we know that VecGCN is beneficial to address the isotropy of directed graph-based GCNs. There are two main reasons: on the one hand, the inclusion of edge vectors makes the importance of neighbor nodes $v$ to the central node $u$ change, and prior to that, the importance is the same 2.2, since the edge vectors are all 1; On the other hand, the order of neighboring nodes cannot be changed during the information aggregation phase, because $e_{ij}x_{ij} \neq e_{ij}x_{ik}$ where $j \neq k$ and $j, k \in \mathcal{N}(i)$, this shows that VecGCN is anisotropic. The pseudo-code for VecGCN is detailed in A.2.

## 4 EXPERIMENTS

In this section, we will verify our proposed VecGCN on 6 datasets: Cora, Citeseer, Pubmed, Texas, Wisconsin, and Cornell(An overview summary of characteristics of the datasets is given in Table~1).

Cora, Citeseer, and Pubmed are popular citation networks with node labels corresponding to scientific subareas and downloaded automatically through the official Deep Graph Library(DGL)[2] program. In these networks, nodes represent papers, and edges denote citations of one paper by another. Node features are the bag-of-words representation of papers, and the node label is the academic topic of a paper.

---

[2]https://github.com/dmlc/dgl

Table 1: Datasets statistics

| Dataset | Cora | Citeseer | Pubmed | Cornell | Texas | Wisconsin |
|---------|------|----------|--------|---------|-------|-----------|
| # Nodes | 2708 | 3327 | 19717 | 183 | 183 | 251 |
| # Edges | 5429 | 4732 | 44338 | 295 | 309 | 499 |
| # Features | 1433 | 3703 | 500 | 1703 | 1703 | 1703 |
| # Classes | 7 | 6 | 3 | 5 | 5 | 5 |

WebKB is a webpage dataset collected from computer science departments of various universities by Carnegie Mellon University and available from the official PyTorch Geometric(PyG)[3] program. We use the three sub-datasets of Cornell, Texas, and Wisconsin, where nodes represent web pages, and edges are hyperlinks between them. Node features are the bag-of-words representation of web pages. The web pages are manually classified into five categories, student, project, course, staff, and faculty.

To guarantee the correctness of our proposed method, we conduct experiments with 12 different GNN models A.1 in the following experiments with pytorch-1.10[4], hardware is an 8-core CPU with 16G Memory, and the GPU is GeForce RTX 2080 (11GB).

### 4.1 TRAINING AND IMPLEMENTATION DETAILS.

#### 4.1.1 NODE CLASSIFICATION

We performed node classification in a semi-supervised setting. Due to the small graph size, we use a 60%/20%/20% training/validation/test split for Cornell, Texas, and Wisconsin. For Cora, CiteSeer, and Pubmed, we use the same split as Kipf & Welling (2016). Because VecGCN is an extension of Kipf & Welling (2016) in higher dimensional space, we take only hyperparameter $d$, which is the dimension of the feature, fixed at 64. The appendixA.5 provides more hyper-parameter details.

#### 4.1.2 LINK EXISTENCE PREDICTION

Given arbitrary two nodes $u$ and $v$, the link existence prediction model is asked to predict if $e_{uv} \in E$, by scoring their inner product of representations $x_u$ and $x_v$ as the likelihood of connectivity, written as $score(e_{uv}) = \langle x_u, x_v \rangle$. We applied negative sampling to VecGCN. Training a link prediction model involves comparing the scores between nodes connected by an edge against the scores between an arbitrary pair of nodes. For example, given an edge connecting $u$ and $v$, we encourage the score between node $u$ and $v$ to be higher than the score between node $u$ and a sampled node $v'$ from an arbitrary noise distribution $v' \sim \mathcal{P}_n(u)$[5], which is $v' \notin \mathcal{N}(u)$. Like the section4.1.1 above, we randomly split the edge-set into 3 parts at each beginning of the epochs: 80% for training, 15% of edges for testing, and 5% for validation. During splitting, the connectivity was maintained. The appendixA.5 provides more hyper-parameter details.

#### 4.1.3 THE IMPACT OF DIMENSION ON VECGCN

The dimension $d$ is the key hyperparameter in VecGCN, which determines the size of the edge direction vector. We tested the effect of dimension $d$ on model performance on two tasks, node classification and link prediction, and three datasets, Cora, Citeseer, and Pubmed. In these experiments, we set the dimension to $2^3$, $2^4$, $2^5$, $2^6$, $2^7$, and $2^8$, respectively.

### 4.2 RESULTS

As shown in Table~2, we see that VecGCN performs well across all tasks, implying that the symmetrization-based approach can benefit citation networks in the context of node classi-

---

[3]https://github.com/pygteam/pytorch_geometric

[4]https://pytorch.org/

[5]$\mathcal{P}_n(v)$ indicates the nodes that are not $u$'s neighbours

fication tasks. This matches our intuition, and if given a paper on the topic of reinforcement learning, then it is likely to cite or to be cited by other reinforcement learning papers.

Table 2: Node classification accuracy (mean±std %). The best results are in bold. Because the best scores are distributed over several models, to make this analysis more quantitative, we compute the absolute difference in the accuracy of each method from that of the top performing method (%) on each data set and take the average as "inferior index", the closer to 0, the better this indicator is. (Digraph and MagNet are out of memory on the pubmed dataset because they use the PyG framework.)

|          | Model   | Cora          | Citeseer        | Pubmed          | Cornell       | Texas         | Wisconsin       | inferior |
|----------|---------|---------------|-----------------|-----------------|---------------|---------------|-----------------|----------|
| Spectral | GCN     | 81.1±0.2      | 70.3±0.4        | **79.4±0.1**    | 59.0±6.4      | 58.7±3.8      | 55.9±5.4        | -14.8    |
|          | JKnet   | **86.1±1.5**  | 70.9±1.9        | 76.6±1.2        | 57.3±5.0      | 61.1±6.2      | 52.8±5.7        | -14.8    |
|          | SGC     | 79.3 ± 0.0    | **71.9±0.1**    | 78.9 ± 0.0      | 58.6±3.4      | 56.4±4.3      | 51.3±6.4        | -16.2    |
|          | GCNII   | 82.2±0.1      | 67.8±0.2        | 78.1±0.1        | 74.2±6.5      | 69.2±6.6      | 70.3±4.8        | -8.6     |
|          | ChebNet | 81.2±0.5      | 69.8±0.2        | 74.4±0.3        | 79.8±5.0      | 79.2±7.5      | 81.6±6.3        | -4.6     |
| Spatial  | GAT     | 82.3±0.3      | 70.0±0.5        | 78.5±0.2        | 57.6±4.9      | 61.1±5.0      | 54.1±4.2        | -15.0    |
|          | SAGE    | 82.1±0.4      | 70.3±0.8        | 78.2±0.4        | 80.0±6.1      | 84.3±5.5      | 83.1±4.8        | -2.6     |
|          | APPNP   | 82.4±0.7      | 70.4±1.1        | 79.3±0.6        | 58.7±4.0      | 57.0±4.8      | 51.8±7.4        | -15.6    |
|          | GIN     | 78.1±2.0      | 63.3±2.5        | 78.9±0.1        | 57.9±5.7      | 65.2±6.5      | 58.2±5.1        | -15.3    |
| Directed | DGCN    | 81.3±1.4      | 66.3±2.0        | 76.9 ± 1.9      | 67.3±4.3      | 71.7±7.4      | 65.5±4.7        | -14.2    |
|          | Digraph | 79.4±1.8      | 62.6±2.2        | -               | 66.8±6.2      | 64.9±8.1      | 59.6±3.8        | -16.1    |
|          | MagNet  | 79.8±2.5      | 67.5±1.8        | -               | 84.3±7.0      | 83.3±6.1      | **85.7±3.2**    | -2.7     |
| Ours     | VecGCN  | 83.1±0.3      | 71.4±0.4        | 79.1±0.2        | **85.7±6.3**  | **84.5±5.1**  | 84.1±3.7        | **-0.9** |

For the node classification task, we added the jump-connection JKnetXu et al. (2018b), the multi-layer stack GCNIIChen et al. (2020), and the low-complexity SGCWu et al. (2019), for detailed descriptions refer to AppendixA.1. The purpose of doing so is to expand the scope of comparison and highlight VecGCN features and advantages. From the above tableTable~2 we can see that as a generalization of vanilla GCN, VecGCN has achieved almost a comprehensive surpass, which shows that VecGCN inherits and carries forward the advantages of GCN. Similarly, the edge vector computation module of VecGCN is also inspired by GAT, which achieves a comprehensive surpass, showing that VecGCN inherits and carries forward the advantages of GAT. In addition, we also see that other directed methods perform relatively poorly on the WebKB network, probably because these graphs are small and have few training samples. Therefore, this does not indicate that those models that perform poorly are inferior. In addition, because of the uneven distribution of optimal scores, to make the analysis more quantitative. We use "inferior" to calculate the difference between each model and the optimal model by calculating the absolute difference between the classification accuracy of each method and that of the best performing method (percentage), and averaging over the six data sets. In this case, a lower score is better, and a method with a score of 0 indicates that the method is the best-performing method on each dataset. As shown in Table~2, VecGCN achieved an overall best result of -0.9%.

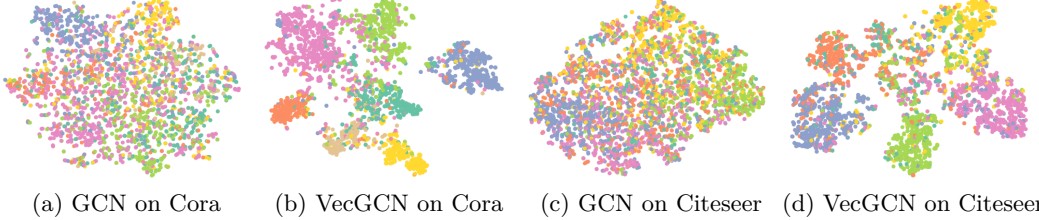

|  (a) GCN on Cora  |  (b) VecGCN on Cora  |  (c) GCN on Citeseer  |  (d) VecGCN on Citeseer  |

Figure 3: t-SNE Analysis: results from visual analysis of the distribution of node representations.

Moreover, we give the visualizationFig~3 of VecGCN on the Cora and Citeseer datasets after dimension reduction on node vectorsVan der Maaten & Hinton (2008). We can see

that VecGCN has a better clustering tendency than GCN. Since GNN can have as sizeable discriminative power as the Weisfeiler-Lehman(WL) testXu et al. (2018a), we argue that this is mainly owed to the fact that VecGCN brings in the edge directional vectors.

In link existence classification task, we achieve the best performance on all datasets in Table~3. Due to the clear clustering tendency of VecGCN, which makes node representations more distinguishable, the discrimination between edge directions is more significant. This matches our intuition and demonstrates that VecGCN is a natural extension of GCN.

Table 3: Link existence classification accuracy (mean±std %). In addition, the spectral domain approach has many jump connections that directly use the information of the adjacency matrix, which is equivalent to label leakage in link existence prediction, and therefore it is not considered for inclusion in this table. (Digraph and MagNet are out of memory on the pubmed dataset because they use the PyG framework.)

|  | Model | Cora | Citeseer | Pubmed | Cornell | Texas | Wisconsin |
|---|---|---|---|---|---|---|---|
| Spectral | GCN | 70.0±0.3 | 65.4±0.8 | 68.9±0.3 | 62.9±1.6 | 63.3±2.6 | 63.7±2.3 |
|  | ChebNet | 70.9±0.5 | 68.2±0.3 | 71.0±0.4 | 70.9±3.2 | 71.6±3.6 | 72.4±2.3 |
| Spatial | GAT | 72.3±0.9 | 68.2±0.4 | 68.4±0.8 | 64.7±1.7 | 64.4±3.2 | 65.5±2.4 |
|  | SAGE | 63.8±0.8 | 62.8±0.6 | 65.1±0.1 | 67.8±3.0 | 66.7±2.6 | 66.4±3.4 |
|  | APPNP | 75.0±0.6 | 71.3±0.8 | 72.6±0.9 | 67.2±2.3 | 67.0±1.8 | 68.1±2.1 |
|  | GIN | 75.4±0.4 | 69.3±0.7 | 72.0±0.8 | 68.2±2.6 | 67.5±3.6 | 69.3±1.6 |
| Directed | DGCN | 80.2±0.4 | 78.2±0.5 | 79.3±0.6 | 77.9±3.4 | 80.2±4.5 | 80.8±2.0 |
|  | Digraph | 80.0±0.2 | 80.3±0.9 | - | 78.7±2.5 | 79.2±2.8 | 80.9±2.6 |
|  | MagNet | 80.7±0.5 | 77.9±0.7 | - | 79.5±3.3 | 79.3±5.3 | 80.8±2.9 |
| Ours | VecGCN | **82.6±1.0** | **81.8±0.2** | **82.1±0.5** | **80.8±5.2** | **81.1±5.1** | **81.2±5.1** |

Finally, we test the effect of dimensions on the model performances. As shown in Fig~A2, a higher dimension is not better, and the model's performance will be saturated. The emergence of this phenomenon will be studied in future works.

## 5 CONCLUSION

In this paper, we propose VecGCN solve the problem of the lack of anisotropy in GNNs. From intuition, we define the edge vectors in Euclidean space. This approach to defining direction at the physical level naturally extends the usability of vanilla GCN. Typical examples include different atoms in a chemical molecule passing messages to each other, relational data in social networks, all kinds of field propagation in crystal lattices, and magnetic anisotropicity in metals and alloys. VecGCN is a new approach to designing spectral-based GCN layers to fit directed graphs without requiring feature vectors. Moreover, this approach can be applied to any GCN model and is compatible with real problems to the maximum extent. Supported by a series of theoretical and empirical results, we believe that our approach will generate new ideas for the study of GCN.

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

## A  APPENDIX - DETAILS

### A.1  LIST OF METHOD ABBREVIATIONS

**Spectral**

1. GCN (Kipf & Welling (2016))
2. JKnet (Xu et al. (2018b))
3. SGC (Wu et al. (2019))
4. GCNII (Chen et al. (2020))
5. ChebNet (Defferrard et al. (2016))

**Spatial**

1. GAT (Veličković et al. (2017))
2. SAGE (Hamilton et al. (2017a))
3. APPNP (Klicpera et al. (2018))
4. GIN (Xu et al. (2018a))

**Directional**

1. DGCN (Zilly et al. (2017))
2. Digraph (Tong et al. (2020a))
3. MagNet (Zhang et al. (2021))
4. VecGCN (this paper)

### A.2  PSEUDO CODE

---
**Algorithm 1** An algorithm with caption
---
$iteration \leftarrow 0$
$\widetilde{L} = (D+I)^{-\frac{1}{2}}(A+I)(D+I)^{-\frac{1}{2}}$
$X_0 = \phi(h) + b$
**while** $iter \leq 200$ **do**
$\quad e_{ij}^l = x_i - x_j$
$\quad \alpha_{ij}^l = (1 - \frac{||e_{ij}^l||_2}{\sum_{j \in \mathcal{N}(i)}(||e_{ij}^l||_2)})/D_i$
$\quad e_{ij}^l = \alpha_{ij}^l e_{ij}^l$
$\quad X^{l+1} = \sigma(\widetilde{L}\alpha^l e^l X^l W^l)$
$\quad iter = iter + 1$
**end while**

---

### A.3  SHIFT INVARIANT

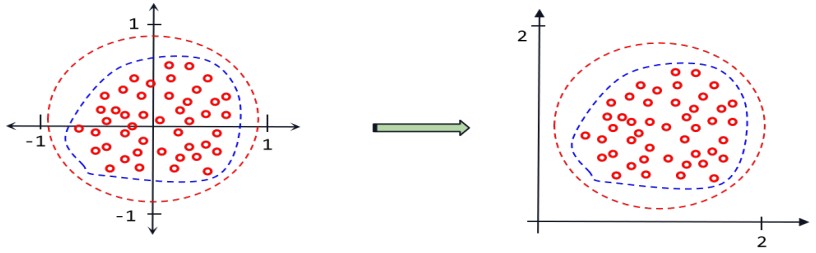

Figure A1: The picture's red points change only in coordinates but not in content.

### A.4  ACCURACY IN DIFFERENT DIMENSIONS

### A.5  HYPER-PARAMETERS DETAILS

## B  APPENDIX - MATHEMATICAL PROOFS

### B.1  VANILLA GCNS

$$X^{l+1} = \sigma(U g_\theta(\Lambda) U^T X^l)$$

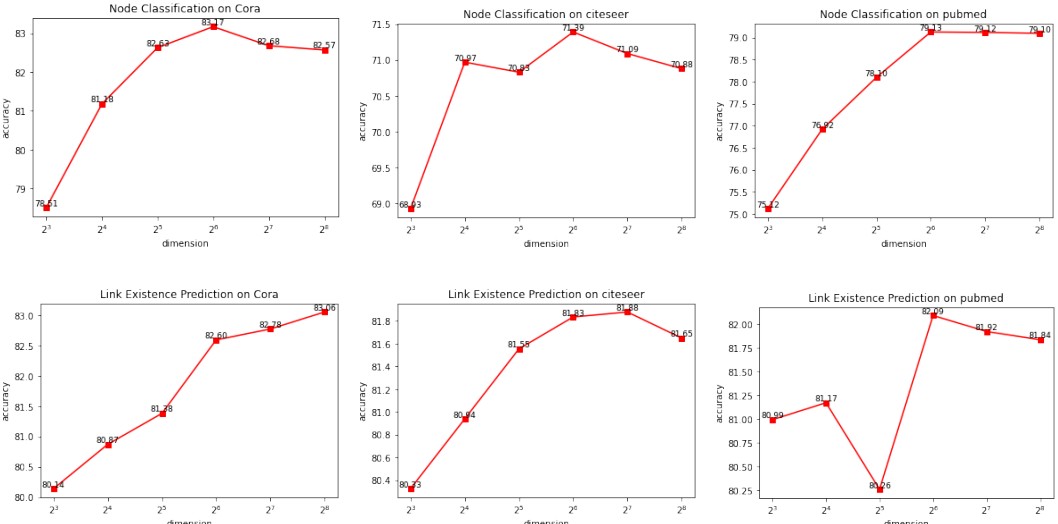

Figure A2: Effect of dimensions(mean %). The vertical axis indicates the accuracy and the horizontal axis the dimension.

Table A1: Hyperparameters for Baseline GNNs

| Hyperparameters | | | | | | |
|---|---|---|---|---|---|---|
| | Models | lr | weight_decay | dropout | hidden | layers | others |
| Spectral | GCN | 0.01 | 5E-4 | 0.5 | 16 | 2 | - |
| | GCNII | 0.01 | 0.01/5E-4 | 0.6 | 64 | 64 | $\lambda$=0.5, $\alpha$=0.2 |
| | SGC | 0.2 | 5E-6 | 0 | - | 1 | - |
| | JKnet | 0.005 | - | 0.5 | 32 | 5 | cat |
| | ChebNet | 0.01 | 5E-4 | 0 | 32 | 2 | K=2,num_hop=2 |
| Spatial | GAT | 0.01 | 5E-4 | 0.6 | 8 | 2 | num_heads=8 |
| | SAGE | 0.001 | 5E-4 | 0.5 | 256 | 3 | mean |
| | APPNP | 0.01 | 5E-4 | 0.5 | 64 | 2 | $\alpha = 0.1$ |
| | GIN | 0.01 | 5E-6 | 0.6 | 64 | 5 | mean |
| Directed | DGCN | 0.001 | 1E-8 | 0.5 | 64 | 5 | cat |
| | Digraph | 0.01 | 1E-4 | 0.5 | 128 | 2 | $\alpha$=0.1 |
| | MagNet | 0.005 | 5E-4 | 0.5 | - | 2 | $K$=1,$q$=0.25 |
| | VecGCN | 0.005 | 5E-4 | 0.5 | 16 | 2 | dimension =64 |

, where $g_\theta(\Lambda) = \sum_{i=0}^{K} \alpha_i \Lambda_i$.

$$Ug_\theta(\Lambda)U^T = U(\sum_{i=0}^{K} \alpha_i \Lambda_i)U^T$$
$$= \sum_{i=0}^{K} \alpha_i U\Lambda_i U^T \qquad \text{(B.1)}$$
$$= \sum_{i=0}^{K} \alpha_i L_i$$

### B.2 LINEAR COMBINATION

*Proof.* The eigenvalues in the extended field $G$ are linear combinations of the eigenvalues in its basis $F$

$\because F \subseteq G$

$\therefore G$ can be viewed as a linear space on $F$, $B \times F \to G$. Thus, $G$ has a group of $F-$basis $\{b_i \in B | i \in \mathcal{I}\}$, where $\mathcal{I}$ is the index set and $B$ is the basis of $G$ on $F$, that is to say, every element in $G$ is a linear combination of $F-$basis $\{b_i \in B | i \in \mathcal{I}\}$. Written as $g_i = b_i \cdot f_i$, where $g \in G, \quad b \in B, \quad f \in F$.

let $M_F = U\Lambda U^T$ be the decomposition of matrix $M$ in $F$.

$$
\begin{aligned}
G &= \vec{B}F \\
M_G &= \vec{B}M_F \\
&= \vec{B}U\Lambda U^T \\
&= U\vec{B}\Lambda U^T
\end{aligned}
\tag{B.2}
$$

From Eq~B.2 we know that, the eigenvalues of $G$ are linear combinations of basis $B$ on $F$ □

### B.3 Value Domain of $\alpha$

*Proof.* Eq~4 Value Domain of $\alpha$

$$
\begin{aligned}
\sum_{j \in \mathcal{N}(i)} (1 - e_{ij}/Z_i) &= \sum_{j \in \mathcal{N}(i)} 1 - \sum_{j \in \mathcal{N}(i)} e_{ij}/Z_i \\
&= D_i - 1
\end{aligned}
\tag{B.3}
$$

bring Eq~B.3 into Eq~4, we have:

$$
\begin{aligned}
\sum_{j \in \mathcal{N}(i)} \alpha_{ij} &= (D_i - 1)/D_i \\
&= 1 - 1/D_i
\end{aligned}
\tag{B.4}
$$

$\therefore \alpha_i \in [0, 1 - 1/D_i] = [0, 1]$ □

### B.4 Field Extension

A subfield $G$ of a field $F$ is a subset $G \subseteq F$ that $G$ is a field with respect to the field operations inherited from $F$. Then $F$ is an extension field of $G$, and $G$ is the basis field of $F$. Such a field extension is denoted $F/G$.

In each field expansion, the subfield can be regarded as a vector space with the base field as the coefficient field. with filed expansion $F/G$, considering the elements in $F$ as vectors and the elements in $G$ as the corresponding coefficients. Under such definition, $F$ is an $G$-vector space. The dimension of the vector space is called the degree of the filed expansion, which is generally denoted as $[F : G]$.

For example, the field of complex numbers $\mathcal{C}$ is an extension field of the field of real numbers $\mathcal{R}$ . Clearly then, $\mathcal{C}/\mathcal{R}$ is a field extension and we have $[\mathcal{C} : \mathcal{R}] = 2$ because $\{1, i\}$ is a basis, such extension $\mathcal{C}/\mathcal{R}$ is finite.

### B.5 Rotation Invariant

*Proof.* Laplacian operator is rotational invariant.

The Laplace operator in GNN is equivalent to that under discrete conditions, where we can prove that the Laplace operator under continuous conditions is rotationally invariant.

let:

$$x = x'cos - y'sin \tag{B.5}$$

$$y = x'sin + y'cos \tag{B.6}$$

where $(x, y)$ are the original coordinates and $(x', y')$ are the rotation coordinates.

$$
\begin{aligned}
\nabla^2 f &= \frac{\partial^2 f}{\partial x'^2} + \frac{\partial^2 f}{\partial y'^2} \\
&= \frac{\partial}{\partial x'}\left(\frac{\partial f}{\partial x'}\right) + \frac{\partial}{\partial y'}\left(\frac{\partial f}{\partial y'}\right) \\
&= \frac{\partial}{\partial x'}\left(\frac{\partial f}{\partial x}\frac{\partial x}{\partial x'} + \frac{\partial f}{\partial y}\frac{\partial y}{\partial x'}\right) + \frac{\partial}{\partial y'}\left(\frac{\partial f}{\partial x}\frac{\partial x}{\partial y'} + \frac{\partial f}{\partial y}\frac{\partial y}{\partial y'}\right) \\
&= \frac{\partial}{\partial x'}\left(\frac{\partial f}{\partial x}\cos\theta + \frac{\partial f}{\partial y}\sin\theta\right) + \frac{\partial}{\partial y'}\left(-\sin\theta\frac{\partial f}{\partial x} + \cos\frac{\partial f}{\partial y}\right) \\
&= \frac{\partial}{\partial x}\left(\frac{\partial f}{\partial x}\cos\theta + \frac{\partial f}{\partial y}\sin\theta\right)\frac{\partial x}{\partial x'} + \frac{\partial}{\partial y}\left(\frac{\partial f}{\partial x}\cos\theta + \frac{\partial f}{\partial y}\sin\theta\right)\frac{\partial y}{\partial x'} \\
&\quad + \frac{\partial}{\partial x}\left(-\sin\theta\frac{\partial f}{\partial x} + \cos\frac{\partial f}{\partial y}\right)\frac{\partial x}{\partial y'} + \frac{\partial}{\partial y}\left(-\sin\theta\frac{\partial f}{\partial x} + \cos\frac{\partial f}{\partial y}\right)\frac{\partial y}{\partial y'} \\
&= \frac{\partial}{\partial x}\left(\frac{\partial f}{\partial x}\cos\theta + \frac{\partial f}{\partial y}\sin\theta\right)\cos\theta + \frac{\partial}{\partial y}\left(\frac{\partial f}{\partial x}\cos\theta + \frac{\partial f}{\partial y}\sin\theta\right)\sin\theta \\
&\quad + \frac{\partial}{\partial x}\left(-\sin\theta\frac{\partial f}{\partial x} + \cos\frac{\partial f}{\partial y}\right)(-\sin\theta) + \frac{\partial}{\partial y}\left(-\sin\theta\frac{\partial f}{\partial x} + \cos\theta\frac{\partial f}{\partial y}\right)\cos\theta \\
&= \frac{\partial}{\partial x}\frac{\partial f}{\partial x} + \frac{\partial}{\partial y}\frac{\partial f}{\partial y} \\
&= \frac{\partial^2 f}{\partial x'^2} + \frac{\partial^2 f}{\partial y'^2}
\end{aligned}
\tag{B.7}
$$

$\square$

### B.6 Proof on the eigenvectors of the skew-symmetric matrix are pure imaginary numbers

*Proof.* The eigenvectors of the skew-symmetric matrix are pure imaginary numbers.

$M$ is a real skew-symmetric matrix of order $T$:

Let $\lambda$ be the eigenvalue of $M$, and $\exists \lambda \in \mathcal{R}$

$$s.t. \quad M\xi = \lambda\xi$$

Multiply both sides by the conjugate transpose of $\xi$, and we have:

$$\overline{\xi^T} M \overline{\xi} = \lambda \overline{\xi^T} \overline{\xi} \tag{B.8}$$

Take the conjugate transpose of both sides: $\overline{\xi^T} M \overline{\xi} = \lambda \overline{\xi^T} \overline{\xi}$

Similarly,

$$\overline{\xi^T}\,\overline{M^T}\,\overline{\xi} = \overline{\lambda}\,\overline{\xi^T}\,\xi \tag{B.9}$$

$$\because \quad \overline{M}^T = -M$$

bring $\overline{M}^T = -M$ into (B.9)

$$-\overline{\xi^T} M \xi = \overline{\lambda}\,\overline{\xi^T}\,\xi \tag{B.10}$$

Add (B.10) to (B.8):

$$(\lambda + \overline{\lambda})\overline{\xi^T}\xi = 0$$

$\therefore \forall \xi \in \mathcal{C}^n , \overline{\xi^T}\xi \neq 0 \Rightarrow \lambda + \overline{\lambda} = 0 \Leftrightarrow real(\lambda) = 0$

$\square$

