# OpenReview forum: "Vectorial Graph Convolutional Networks"
_ICLR.cc/2023/Conference — Submitted to ICLR 2023_

### Official Review · Reviewer_opRy · 2022-10-24

**Confidence:** 5
**Correctness:** 3
**Technical Novelty And Significance:** 2
**Empirical Novelty And Significance:** 2
**Recommendation:** 3

**Clarity, Quality, Novelty And Reproducibility:**

The clarity and quality of this paper are good, considering its clear expressions and experiments results, while the novelty is one of my biggest concerns. It would be good if the authors can address these concerns above in the rebuttal.

**Details Of Ethics Concerns:**

No ethics concerns.

**Strength And Weaknesses:**

**Strength:**

This paper has some advantages as follows:
- This paper focuses on a fundamental and important research problem namely how to generalize the spectral convolution operator to directed graphs.

- The paper is clearly written and the proposed method is easy to understand. Figure 2 helps readers to better understand the calculation mechanisms which is good.

- The experiments on link classification tasks well verify the effectiveness of the proposed VecGCN, which achieves consistent improvements against baselines.

**Weaknesses:**

However, I still have some concerns:
- The main concern is about the technical novelty. I think the technique of replacing the adjacency matrix with the distance matrix has been explored in the literature, e.g. DMPNN [1] and the directed graph neural networks have also been proposed in the community, e.g. R-GCN [2]. Moreover, I do not agree that “GCNs still limited to the undirected graph”. In a narrow sense, this condition of an undirected graph  is only required by the spectral interpretation  of GCNs. The general framework --  message passing paradigm can deal with the directed graph smoothly.

- Technically, I think that using distance matrix to replace the adjacency matrix just evades the asymmetry problem of directed graph, rather than solving it. I can't see any theoretical justification of this replacement.

- The time complexity as well as the empirical efficiency of the proposed VecGCN is not well discussed, since relatively complex techniques are proposed in the model.

- I also have concerns about the experiments. The improvements of the VecGCN seem not significant compared with baselines in node classification tasks. Furthermore, it would be better to consider Open Graph Benchmark in the experiments for more convincing results.

[1] Yang, Kevin, et al. "Analyzing learned molecular representations for property prediction." Journal of chemical information and modeling 59.8 (2019): 3370-3388.

[2] Schlichtkrull, Michael, et al. "Modeling relational data with graph convolutional networks." European semantic web conference. Springer, Cham, 2018.


**Summary Of The Paper:**

This paper focuses on the problem that graph convolutional networks are limited to the undirected graph due to theoretically needing the symmetric matrix for the Laplacian transform. To tackle the problem, this paper generalizes the spectral convolution operator to directed graphs by field extension, overcoming the symmetric problem as well as the information loss problem. Some experimental results show the effectiveness of the proposed VecGCN on several directed graph node classification and link prediction tasks.

**Summary Of The Review:**

In summary, this paper gives an alternative construction of GCNs on directed graph. The main concerns lie in the technical novelty and some experiment analyses. I vote reject for this paper.

---

### Official Review · Reviewer_T4f3 · 2022-10-24

**Confidence:** 4
**Correctness:** 3
**Technical Novelty And Significance:** 2
**Empirical Novelty And Significance:** 2
**Recommendation:** 1

**Clarity, Quality, Novelty And Reproducibility:**

As discussed above, the paper is generally not well-written and it is hard to read. The authors authors did not submit the implementation of the proposed approach, thus it might be hard for the reader to reproduce the reported results. The quality of the paper is poor and as discussed above there are several weak points. The novelty of the proposed method is limited, and the authors do not explain what are its advantages over previous methods.

**Strength And Weaknesses:**

Strengths:

- The paper deals with a limitation of spectral GNNs which cannot be directly applied to directed graphs. The proposed method is relatively simple which a good thing in my opinion, however, as discussed below, there are several issues that the authors need to address.

- The proposed approach seems to perform relatively well in the two considered tasks. In the task of link prediction, it outperforms all the baselines on all datasets, while in node classification, it is the best-performing approach on 2 out of the 6 datasets.

Weaknesses:

- Most existing message passing graph neural networks can directly handle directed graphs (since they operate on the adjacency matrix which is not assumed to be symmetric), while there have also been proposed several models for that kind of graphs. Thus, the authors should explain in more details what makes the proposed approach appropriate for processing directed graphs and what are its advantages over previous models. Extensive experimental analysis and mathematical proofs of the soundness of the proposed method are rather limited.

- Several details of the proposed approach are not properly explained in the paper. For instance:
	- In subsection 3.1, how exactly is the distance between two nodes defined? In my understanding, it is the Euclidean distance between the nodes' representations. By replacing the adjacency matrix with the matrix of distances between nodes, the graph's structural information is lost. I would like the authors to comment on that. Furthermore, the matrix of distances is likely to be dense (distances are positive in most cases). This is another limitation of the method since it implies that the proposed approach cannot be applied to very large graphs since the distance matrix might not fit in memory.
	- It is not clear to me why the scalar elements of the distance matrix are mapped to a high dimensional space. I suggest the authors provide clear explanations.

- In the experimental evaluation, several benchmark datasets are not considered (especially larger ones such as ogbn-arxiv, squirrel, actor, Amazon Computers, Amazon Photo for the node classification task). It is thus not clear whether the proposed approach can be applied to large datasets.

- The writing is not very clear. In fact, the paper leaves the reader with a feeling that this was an initial draft of the work. I found it relatively difficult to follow the analysis due to multiple grammatical and syntactic mistakes, while the vocabulary used is in many cases poor. Therefore, the considered manuscript lacks clarity and I suggest the authors work on improving that.

**Summary Of The Paper:**

This paper deals with the problem of the GCN architecture being limited to undirected graphs due to the requirement of the symmetric matrix as the basis for the Laplacian transformation. To overcome this limitation the authors propose (i) replacing the adjacency matrix with the distance matrix and (ii) extending the elements of the adjacency matrix from scalars to vectors using the field extension method to include the concept of direction. They evaluate the proposed model on six benchmark datasets for node classification and link prediction tasks and compare the performance to several baselines.

**Summary Of The Review:**

In my view, the paper seems to be proposing an incremental contribution for the graph representation learning community. Due to reasons discussed above (related to limited novelty, poor presentation, and incomplete experimental evaluation), I think the work in its current state is not quite ready for publication, thus I cannot recommend acceptance.

---

### Official Review · Reviewer_DTdX · 2022-10-24

**Confidence:** 4
**Correctness:** 2
**Technical Novelty And Significance:** 2
**Empirical Novelty And Significance:** 2
**Recommendation:** 1

**Clarity, Quality, Novelty And Reproducibility:**

Clarity: The theoretical analysis of the motivation of VecGCN needs further clarification.

Quality: The writing is poor and the manuscript seems incomplete.

Novelty: The idea of generalizing GNNs on directed graphs with filed extension is relatively novel.

Reproducibility: On one hand, the details of the experimental setup are satisfactory. No codes are available on the other hand weak reproducibility.

**Strength And Weaknesses:**

Strengths:
* The idea of generalizing GCNs from undirected graphs to directed ones is of great importance in some practical scenarios.
* The use of field extension to regard the graph as the basis of its extension to define the edge vector space is interesting.

Weaknesses:
* This paper is a bit loosely written as it contains many typos, grammar errors, and incomplete sentences. I suggest the authors do thorough proofreading during the rebuttal. For examples: Eq~B.10 -> Eq~2 in line 1, page 6; adjacent matrix -> adjacency matrix in line 1 of Sec. 3.3. Sections A.3, A.4 and A.5 are all empty.
* The novelty of this paper is limited, especially the theoretical parts. The theoretical proof of B.5 that the Laplacian operator is rotational invariant is relatively a well-known property of the Laplacian operator. The theoretical proof of B.6 that the eigenvectors of the skew-symmetric matrix are pure imaginary numbers seems to be disconnected from the main context as well as the design of VecGCN, which is strange to be included in the Appendix.
* For experiments, the concerns include:
    - The chosen benchmarks are all undirected graphs, while VecGCN is designed with a clear motivation for directed graphs. It is unclear why the performance of VecGCN on undirected graphs could demonstrate its effectiveness of VecGCN. Moreover, VecGCN performs worse on 4 out of 6 datasets in the node classification task, which makes VecGCN even weak.
    - Most graphs in the benchmarks seem to be small. The scalability of VecGCN is unclear and needs to be demonstrated with some massive graphs such as benchmarks from OGB.
    - For the link prediction task, some GNNs that were deliberately designed for this task are missing, such as [1-2]. The authors might include them as baselines for comparison for better demonstration. Moreover, the link direction prediction task is more suitable and should be included.

[1] Link prediction based on graph neural networks, NeurIPS 2018

[2] Line graph neural networks for link prediction, TPAMI 2021


**Summary Of The Paper:**

This paper proposes a new variant of the GCN model, namely VecGCN, based on replacing the adjacency matrix as the distance matrix. The main idea of VecGCN is to solve the requirement for using a symmetric matrix in GCNs as the basis of Fourier transformation. The authors propose to define edge vectors with the help of filed extension on graphs to denote the direction. They further use the edge vectors to compute the weights of edges for aggregation in a GAT-like style.
Experiments on both node classification and link prediction tasks over six graph benchmark datasets show that VecGCN could perform on par with many spectral-based GNNs.


**Summary Of The Review:**

My concerns are mainly from two aspects:
* The current version seems to be incomplete and the theoretical analysis seems to be redundant.
* The experimental analysis is insufficient and can be further improved to make it more convincing.

---

### Decision · Program_Chairs · 2023-01-20

**Decision:**

Reject

**Justification For Why Not Higher Score:**

No rebuttal to address reviewers' concerns

**Justification For Why Not Lower Score:**

N/A

**Metareview: Summary, Strengths And Weaknesses:**

The paper proposes a new GNN architecture (VecGCN) extending the graph convolutional model to directed graphs in order to take advantage of directionality. Directed graphs have asymmetric adjacency/laplacian matrices making it impossible to use straightforward spectral analysis.

Reviewers had concerns about the novelty and evaluation and initially gave low scores and several questions for the rebuttal. No rebuttal was provided by the authors. The AC recommends rejection.